# MeshWeaver: Sparse-Voxel-Guided Surface Weaving for Autoregressive Mesh Generation

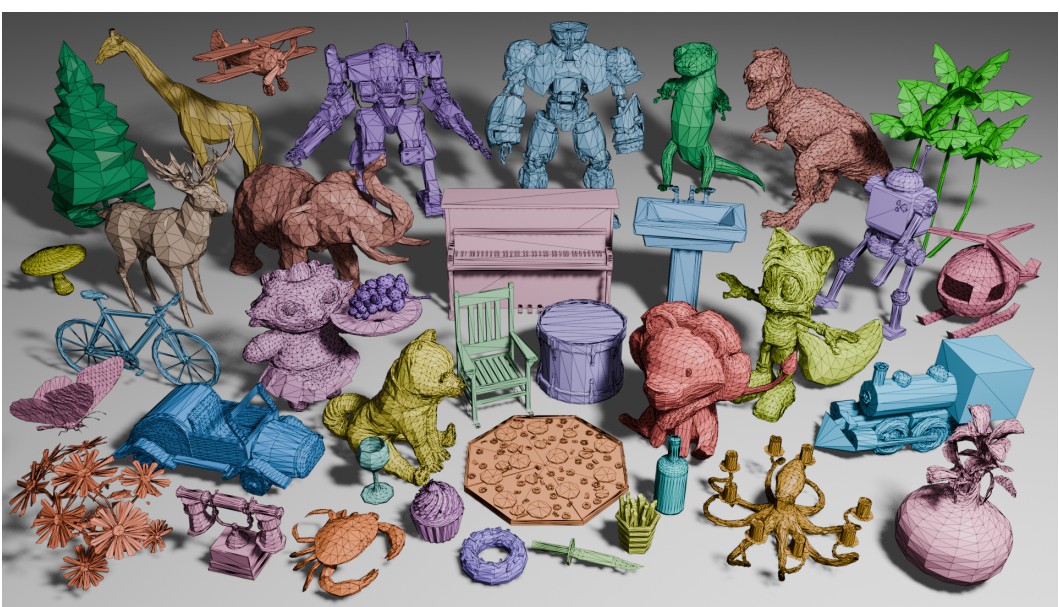

Figure 1: MeshWeaver generates high-quality 3D meshes autoregressively with a sparse-voxel-guided surface weaving process. By directly predict next vertices instead of coordinates, it achieves a state-of-the-art mesh compression ratio of 18%, and can generate meshes with up to 16K faces.

## Abstract

Autoregressive mesh generation has gained attention by tokenizing meshes into sequences and training models in a language-modeling fashion. However, existing approaches suffer from two fundamental limitations: (i) low tokenization efficiency, which yields long sequences and prevents scaling to high-poly meshes, and (ii) absence of geometry-aware guidance, as generation is conditioned only on global shape embeddings rather than local surface cues. We introduce **Mesh-Weaver**, an autoregressive framework that treats mesh generation as a surface weaving process by directly predicting the next vertex instead of independent coordinates. At its core is a multi-level sparse-voxel encoder that injects geometric context into the generative process in three complementary ways: providing voxel features as vertex representations, guiding token prediction via cross-attention to voxel features, and serving as a structural scaffold that constrains generation around the input surface. Our hierarchical design enables coarse-to-fine vertex prediction in a single decoding step, while tightly couples the generative model with 3D geometry. Extensive experiments demonstrate that MeshWeaver achieves a state-of-the-art compression ratio of 18%, can generates meshes with up to 16K faces, and significantly improves geometric fidelity over prior approaches.

## 1 Introduction

Polygonal meshes remain a cornerstone representation of 3D geometry, underpinning applications ranging from games and animation to simulation and virtual reality. But their irregular structure

makes them difficult to model with deep generative architectures. Recent advances therefore rely on implicit representations with mesh extraction via Marching Cubes (Lorensen & Cline, 1998), which eases learning but often produces overly dense, topologically complex meshes that hinder editing and deformation. In contrast, artist-created meshes are carefully crafted to maintain clean topology that facilitates practical usage, yet producing such meshes manually is notoriously labor-intensive. These limitations highlight the importance of automatic mesh generation, which seeks to unite the structural advantages of handcrafted meshes with the scalability of modern generative models.

Recent advances have established autoregressive modeling as a new paradigm for mesh generation. Early attempts such as MeshGPT (Siddiqui et al., 2024) and MeshXL (Chen et al., 2024a) demonstrated the feasibility of tokenizing faces into discrete coordinate sequences and modeling them with transformers, but they suffered from long token sequences and limited scalability to high-poly meshes. Follow-up works explored more compact tokenizations: EdgeRunner (Tang et al., 2025) and TreeMeshGPT (Lionar et al., 2025) leverage half-edge structures for efficient face traversal, while BPT (Weng et al., 2025) and DeepMesh (Zhao et al., 2025a) employ block-wise indexing to reduce coordinate counts. Nevertheless, the predominant next-coordinate prediction paradigm still suffers from two fundamental limitations: (i) producing long token sequences that burden training and inference of autoregressive transformers, and (ii) the generative process depends on global shape embeddings and static vocabulary representations, offering little integration of local geometric context, making it challenging to preserve fine-grained surface fidelity in generated meshes.

To address these challenges, we propose **MeshWeaver**, an autoregressive mesh generation framework that formulates the task as a *surface weaving* process. While prior autoregressive methods also incorporate geometric conditions such as point clouds, they predominantly interpret the task as *conditional shape generation*. In contrast, we advocate a different perspective: the autoregressive paradigm is most effective when posed as a *re-topology* method under known geometry. Compared to 3D generation models based on implicit representations (Xiang et al., 2025; Zhao et al., 2025b), its distinct strength lies in directly producing structured polygonal meshes without relying on post-hoc surface extraction. By shifting the focus to topology construction conditioned on the input surface, we can inject fine-grained geometric priors into every prediction step, guiding the weaving process toward meshes that are both structurally coherent and faithful to the underlying geometry.

MeshWeaver shifts the mesh generation paradigm from next-coordinate to next-vertex prediction. Instead of expending model computation on every independent coordinate, the model directly predicts vertices as atomic tokens in a multi-level coarse-to-fine manner within a single decoding step. This reduces sequence length and allows the transformer to focus on structural reasoning rather than redundant coordinate generation. Central to this design is a **hierarchical sparse-voxel encoder** that injects local geometric context into the autoregressive generation process through three complementary mechanisms: providing multi-level voxel features as vertex representations, guiding token prediction via spatial-aware cross-attention, and serving as a structural scaffold that constrains generation around the input surface. Through this synergy, MeshWeaver surpasses prior limits, achieves a state-of-the-art compression ratio of 18%, generates meshes with up to 16K faces, and delivers significant improvements in geometric fidelity. Our contributions can be summarized as:

- We propose MeshWeaver, an autoregressive framework that formulates mesh generation as a *surface weaving* process, shifting the generation paradigm from next-coordinate to next-vertex prediction for shorter sequences and stronger structural reasoning.

- We design a hierarchical sparse-voxel encoder that injects fine-grained geometric guidance into the generation process at three levels—representation, token prediction, and scaffolding—enabling coherent and geometry-faithful mesh construction.

- MeshWeaver achieves a state-of-the-art mesh compression ratio of 18%, scales to meshes with up to 16K faces, and substantially improves geometric fidelity.

## 2 RELATED WORK

**3D Generation.** Early 3D generation methods (Poole et al., 2023; Wang et al., 2023; Xu et al., 2023; Chen et al., 2023; Tang et al., 2023) adapted 2D models via optimization but were inefficient and produced impractical results. With large-scale 3D datasets (Deitke et al., 2023b;a), recent works follow a "VAE + latent diffusion" paradigm: VecSet representations (Zhang et al., 2023; 2024;

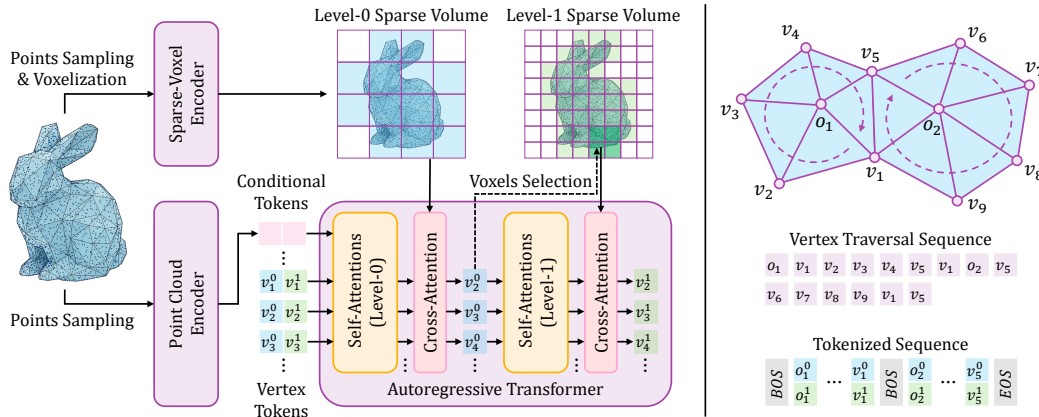

Figure 2: **Left: Overall Pipeline of MeshWeaver.** Given an input surface, we voxelize it and sample points to extract multi-level features with a sparse-voxel encoder. These features provide geometry-aware context that (i) represent vertices, (ii) guide token predictions via cross-attention, and (iii) act as a generation scaffold. The transformer autoregressively weaves the mesh vertex by vertex in a coarse-to-fine manner, attending to voxel features for local geometric context. **Right: Vertex-Level Mesh Tokenization.** The mesh is traversed patch-by-patch to produce compact 2D vertex tokens, greatly shortening sequences.

Wu et al., 2024; Li et al., 2025a;c; Zhao et al., 2025b; Chen et al., 2025a; Li et al., 2025b) yield compact and transferable shape sets but lack fine-grained detail, while sparse-voxel methods (Ren et al., 2024; Xiang et al., 2025; Wu et al., 2025; He et al., 2025; Li et al., 2025d; Chen et al., 2025c) capture local geometry more faithfully but require heavier training. However, both directions focus only on geometry and rely on post-processing (e.g., Marching Cubes), often producing overly dense meshes that limit practical applications.

**Mesh Re-topology.** Mesh re-topology converts raw or high-resolution surfaces into clean, low-poly meshes with consistent topology, which is essential for editing, animation, and texture mapping. In practice, this is still largely done manually, making it costly and skill-intensive. Classical algorithms such as surface simplification (Garland & Heckbert, 1997), quad remeshing (Bommes et al., 2009; Huang et al., 2018), and parameterization methods (Floater & Hormann, 2005) reduce effort but depend on heuristics and are computationally heavy. Recent learning-based methods (Potamias et al., 2022; Dong et al., 2025b;a; Zhang et al., 2025) offer progress, yet re-topology remains challenging due to the need to balance fidelity and compactness while producing workflow-ready meshes.

**Autoregressive Mesh Generation.** PolyGen (Nash et al., 2020) pioneered an autoregressive approach that generated ordered vertex sequences and then connected them into faces with two autoregressive transformers. Subsequent methods such as MeshGPT (Siddiqui et al., 2024) and MeshXL (Chen et al., 2024a) discretized faces into token sequences but suffered from extremely long streams, limiting scalability. To improve compression, later works explored (i) *topology-aware traversal*, which maximizes edge sharing (Chen et al., 2024b; 2025b; Tang et al., 2025; Lionar et al., 2025) or decomposes meshes into local patches to reduce redundant tokens (Weng et al., 2025; Wang et al., 2025b); and (ii) *block-wise coordinate compression*, which partitions space and encodes each vertex by block and offset indices, merging repeated block codes for higher compression (Weng et al., 2025). In parallel, architectural innovations such as hourglass Transformers (Hao et al., 2024), linear-attention mechanisms (Wang et al., 2025a), and reinforcement-learning strategies (Zhao et al., 2025a; Liu et al., 2025) have been explored. Nevertheless, the state-of-the-art compression ratio of mesh tokenization remains capped at about 22%, and mainstream approaches still rely on next-coordinate prediction without explicit local geometric guidance.

## 3 METHOD

### 3.1 PRELIMINARY: MESH TOKENIZATION

A triangle mesh consists of a collection of faces $\mathcal{M} = \{\boldsymbol{f}_1, \boldsymbol{f}_2, \ldots, \boldsymbol{f}_N\}$, where each face is a triplet of vertices $\boldsymbol{f}_i = (\boldsymbol{v}_{i1}, \boldsymbol{v}_{i2}, \boldsymbol{v}_{i3})$, and each vertex is represented by 3D coordinates $\boldsymbol{v}_j = (v_j^x, v_j^y, v_j^z)$.

Unlike textual data, mesh tokenization is considerably harder due to spatial redundancy and irregular connectivity. The most naïve mesh tokenization is to flatten all vertex coordinates into a sequence:

$$\mathcal{M} = \{v_1^x, v_1^y, v_1^z, \ldots, v_{3N}^x, v_{3N}^y, v_{3N}^z\}, \tag{1}$$

where vertices and faces are sorted in some order (*e.g.*, $yzx$-order) and coordinates are discretized into a finite resolution grid (*e.g.*, 7-bit quantization in a $128^3$ grid). In autoregressive mesh generation, the mesh is then modeled as a sequence of tokens, with each coordinate predicted conditional on its predecessors: $p(\mathcal{M}) = \prod_{t=1}^{9N} p(c_t \mid c_{<t})$, where $c_t$ denotes the $t$-th coordinate token.

However, this naïve formulation yields extremely long sequences ($9N$ tokens for a mesh with $N$ faces), severely limiting scalability. To improve compression ratio, later works pursued more compact tokenizations. Topology-aware traversals (Chen et al., 2025b; Tang et al., 2025; Lionar et al., 2025) reduce redundant vertices by maximizing edge sharing, while patch-based methods (Weng et al., 2025; Wang et al., 2025b; Zhao et al., 2025a) shorten sequences via local patch grouping and block-wise coordinate compression. Despite these advances, coordinate-level tokenization remains capped at about 22% compression, leaving the quest for more compact yet faithful tokenization an open challenge.

## 3.2 VERTEX-LEVEL MESH TOKENIZATION

To overcome the compression bottleneck of coordinate-level tokenization schemes, we propose *vertex-level tokenization*, which elevates the basic modeling unit from coordinates to vertices. The key insight is that mesh traversal naturally operates on vertices: the traversal process can be viewed as "weaving" the mesh surface vertex by vertex, akin to threading along the manifold to reconstruct topology. Based on this perspective, we lift the 1D coordinate sequence into a 2D vertex sequence and reformulate the task from next-coordinate prediction to *next-vertex prediction*. In each decoding step, the transformer directly predicts a complete vertex rather than an individual coordinate. This design fully leverages the model's sequence modeling capacity, significantly enhances mesh generation efficiency.

**Mesh Patchification.** The notion of "lifting" tokenization to vertex-level is orthogonal to the mesh traversal strategy and can be integrated with various traversal algorithms. In this work, we adopt a patch-based traversal due to its inherent locality, high efficiency, and minimal reliance on auxiliary tokens. Specifically, we follow the heuristic introduced in BPT (Weng et al., 2025): we begin with all sorted faces marked unvisited, pick the first unvisited face, and identify its vertex connected to the largest number of remaining unvisited faces as the patch center. The patch is then formed by grouping this center with all incident faces. As Figure 2 (right) shows, the mesh is divided into a sequence of $P$ local patches, each consisting of a center vertex $\boldsymbol{o}_i$ and its surrounding vertices $\boldsymbol{v}_{ij}$ arranged in a clockwise manner:

$$\mathcal{M} = \{\boldsymbol{o}_1, \boldsymbol{v}_{11}, \ldots, \boldsymbol{o}_2, \boldsymbol{v}_{21}, \ldots, \ldots, \boldsymbol{o}_P, \boldsymbol{v}_{P1}, \ldots\}. \tag{2}$$

**Multi-Level Vertex Representation.** A crucial challenge in vertex-based tokenization is how to generate a complete vertex within a single decoding step. Prior attempts such as TreeMeshGPT adopt hierarchical MLP heads to sequentially predict $z$, $y$, and $x$ coordinates: $p(\boldsymbol{v}_i) = p(v_i^z) \cdot p(v_i^y \mid v_i^z) \cdot p(v_i^x \mid v_i^z, v_i^y)$. However, the three coordinates of a vertex are strongly coupled and do not exhibit a clear sequential dependency, making such factorization suboptimal.

Instead, we adopt a *multi-level vertex representation* inspired by block-wise indexing (Weng et al., 2025). The 3D space is hierarchically partitioned into voxel grids at $L$ levels. At the $l$-th level, we divide the voxel grids by a factor of $D_l$, leading to a finest resolution of $R = \prod_{l=0}^{L-1} D_l$ that equals to the coordinate quantization resolution. Each voxel at level $l-1$ corresponds to a $D_l^3$ subvolume at level $l$, and each vertex is represented by multi-level voxel indices: $\boldsymbol{v}_i = (v_i^0, \ldots, v_i^{L-1})$, where $v_i^l \in [0, \ldots, D_l^3 - 1]$ denotes the index at level $l$ conditioned on its parent in level $l-1$. The decoding process at step $j$ follows a coarse-to-fine voxel refinement: $p(\boldsymbol{v}_j) = \prod_{l=0}^{L-1} p(v_j^l \mid v_j^{<l})$, which first determines a coarse voxel and progressively narrows the prediction to finer subvolumes until the final resolution is reached.

Figure 3: **Network Architectures.** Left: sparse-voxel encoder. Right: autoregressive transformer.

**Tokenization Results.** By integrating patch-based mesh traversal with multi-level vertex representation, we obtain a 2D vertex-token sequence:

$$\mathcal{M} = \left\{ \begin{bmatrix} \text{BOS} \\ \vdots \\ \text{BOS} \end{bmatrix}, \begin{bmatrix} o_1^0 \\ \vdots \\ o_1^{L-1} \end{bmatrix}, \begin{bmatrix} v_{11}^0 \\ \vdots \\ v_{11}^{L-1} \end{bmatrix}, \dots, \begin{bmatrix} \text{BOS} \\ \vdots \\ \text{BOS} \end{bmatrix}, \begin{bmatrix} o_P^0 \\ \vdots \\ o_P^{L-1} \end{bmatrix}, \begin{bmatrix} v_{P1}^0 \\ \vdots \\ v_{P1}^{L-1} \end{bmatrix}, \dots, \begin{bmatrix} \text{EOS} \\ \vdots \\ \text{EOS} \end{bmatrix} \right\}. \tag{3}$$

Here, a BOS token is inserted at the beginning of each patch to explicitly distinguish the patch center from other vertices, while an EOS token terminates the full sequence. This design yields a compression ratio of 18%, establishing a new state of the art.

## 3.3 SPARSE-VOXEL-GUIDED AUTOREGRESSIVE MESH GENERATION

Previous autoregressive mesh generation approaches typically recast the task as point cloud conditioned coordinate prediction. The input point cloud is encoded into global shape embeddings and then injected into the transformer via prefix tokens or cross-attention. During generation, each coordinate token is represented by a static vocabulary embedding, and the next token is directly predicted from the last-layer hidden state. This paradigm lacks fine-grained structural cues, struggles to faithfully capture the underlying geometry, as it lacks fine-grained structural cues that can guide generation toward high-fidelity surface reconstruction.

To inject fine-grained geometric information and achieve higher-fidelity mesh generation, we introduce a sparse-voxel encoder into the autoregressive generation framework that encodes the input surface into hierarchical voxel features. It enhances the generation pipeline from 3 aspects: (i) each input vertex is represented with multi-level voxel features carrying rich geometric information instead of shape-agnostic static vocabulary embeddings, (ii) before predicting each level of a vertex token, the hidden state attends to corresponding sparse-voxel features to perceive local geometry and adaptively refine predictions, (iii) the sparse voxels themselves provide explicit spatial anchors of the surface, effectively constraining the vertex prediction to regions near the true geometry.

**Sparse-Voxel Encoder.** Given a mesh $\mathcal{M}$, we first voxelize its surface at resolution $R$ to obtain non-empty sparse voxels, and sample a point cloud with normals $\{\boldsymbol{p}_i \in \mathbb{R}^6\}_{i=1}^{N_p}$. As shown in Figure 3, a lightweight PointNet (Qi et al., 2017) aggregates the points inside each voxel into a feature vector. These per-voxel features, together with their voxel coordinates, are processed by a stack of shifted-window sparse attention layers (Weng et al., 2025) to produce sparse voxel features at resolution $R$. To capture multi-scale context, we apply successive sparse convolutional down-sampling layers interleaved with sparse attention, halving the spatial resolution at each stage until reaching the coarsest level 0 of resolution $D_0$. The encoder thus yields a hierarchy of sparse voxel features:

$$\mathcal{F} = \{\mathbf{F}^0, \mathbf{F}^1, \dots, \mathbf{F}^{L-1}\}, \tag{4}$$

where $\mathbf{F}^l \in \mathbb{R}^{N_l \times C_l}$ denotes features of the sparse voxels at level $l$.

**Voxel Features as Vertex Representation.** In our next-vertex prediction paradigm, the transformer operates on vertices represented not by static embeddings but by geometry-aware voxel features. Since each vertex $\boldsymbol{v}_i = (v_i^0, \dots, v_i^{L-1})$ corresponds to voxel indices across levels, we retrieve the features of the associated voxels and concatenate them into a multi-level embedding:

$$\mathbf{e}(\boldsymbol{v}_i) = \text{Concat}\big(\mathbf{F}^0[v_i^0], \mathbf{F}^1[v_i^1], \dots, \mathbf{F}^{L-1}[v_i^{L-1}]\big). \tag{5}$$

This shape-dependent representation encodes rich local geometry around the vertex, substantially enhancing the expressiveness compared to shape-agnostic vocabulary embeddings.

**Cross-Attention-Guided Token Prediction.** Our autoregressive decoder adopts a multi-level structure that mirrors the hierarchical vertex representation. Each level consists of self-attention layers followed by a prediction head. The hidden states and voxel prediction from level $l-1$ are concatenated and linearly projected to condition level $l$ prediction, thus modeling coarse-to-fine refinement. To further inject geometric priors, each prediction head integrates a cross-attention layer: the hidden states serve as queries, while level-$l$ sparse voxel features act as keys and values. The output is passed to a linear layer to predict a $D_l^3$-dimensional distribution over voxels (for level 0, we add BOS and EOS tokens, yielding $D_0^3+2$ classes). For $l > 0$, the voxel predicted at the previous level localizes a subvolume in level $l$, and cross-attention is restricted to voxels inside that subvolume, greatly reducing computation while preserving spatial precision.

**Sparse Voxels as Generation Scaffold.** Unlike prior autoregressive approaches that rely on implicit shape embeddings and risk drifting into empty space, our sparse-voxel representation explicitly marks the occupied regions across different resolutions. During decoding, we leverage this property by masking out probabilities of empty voxels in the prediction head. Concretely, for the $D_l^3$ output distribution at level $l$, only non-empty voxels are retained while the rest are assigned $-\infty$ before sampling. This ensures that every predicted vertex remains anchored to the surface, providing a reliable scaffold that enforces geometric validity throughout the generation process.

### 3.4 TRAINING AND INFERENCE DETAILS

**Training-time Subvolume Pruning.** As described before, when predicting a level-$l$ token ($l > 0$), cross-attention is restricted to the sparse-voxel features located within the subvolume identified by the previous level. During training, however, computing cross-attention over the full mesh sequence requires each vertex to be individually masked to its corresponding subvolume—a process that remains computationally expensive despite the inherent sparsity of the mask. To further reduce training complexity, we introduce a *subvolume pruning* strategy. As Figure 4 shows, since the sparse voxels are naturally partitioned by subvolumes, we sample only a subset of these subvolumes along with the vertices that attend to them, and compute the loss exclusively within this subset. This truncated training significantly decreases the number of sparse voxels involved in cross-attention, thereby accelerating training.

**Cross-Attention KV Cache.** Key–Value (KV) caching is widely adopted in LLM inference to avoid redundant computation. In our model, caching applies not only to the self-attention layers of the autoregressive transformer, but also to the cross-attention inside each prediction head.

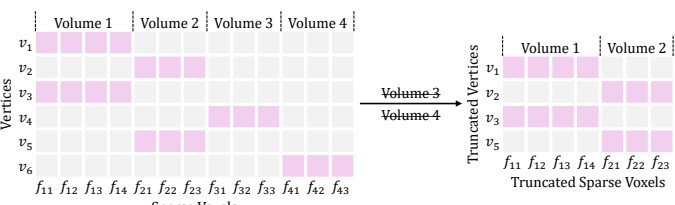

Figure 4: **Training-time Subvolume Pruning.**

After the sparse-voxel encoder produces multi-level voxel features, we map them once into keys and values and store them in a dedicated cross-attention cache. During decoding, the prediction result from the previous level determines a subvolume in current level, and the model retrieves only the relevant sparse keys and values from the cache for prediction. This mechanism eliminates repeated feature projections, substantially reducing inference cost without sacrificing accuracy.

## 4 EXPERIMENTS

### 4.1 EXPERIMENTAL SETTINGS

**Implementation Details.** We build a corpus of 800K meshes by merging Objaverse++(Deitke et al., 2023a), ShapeNet (Chang et al., 2015), 3D-Future (Fu et al., 2021), HSSD (Khanna* et al., 2023), and ABO (Collins et al., 2022), filtering meshes with 1K–16K faces and applying random scale/rotation augmentations. The backbone is a 24-layer LLaMA3-style (Dubey et al., 2024) transformer (1024 hidden, RoPE) with a sparse-voxel and point-cloud encoder, totaling 600M parameters. Coordinates are 7-bit quantized with a two-level space partition ([16, 8]). Training uses

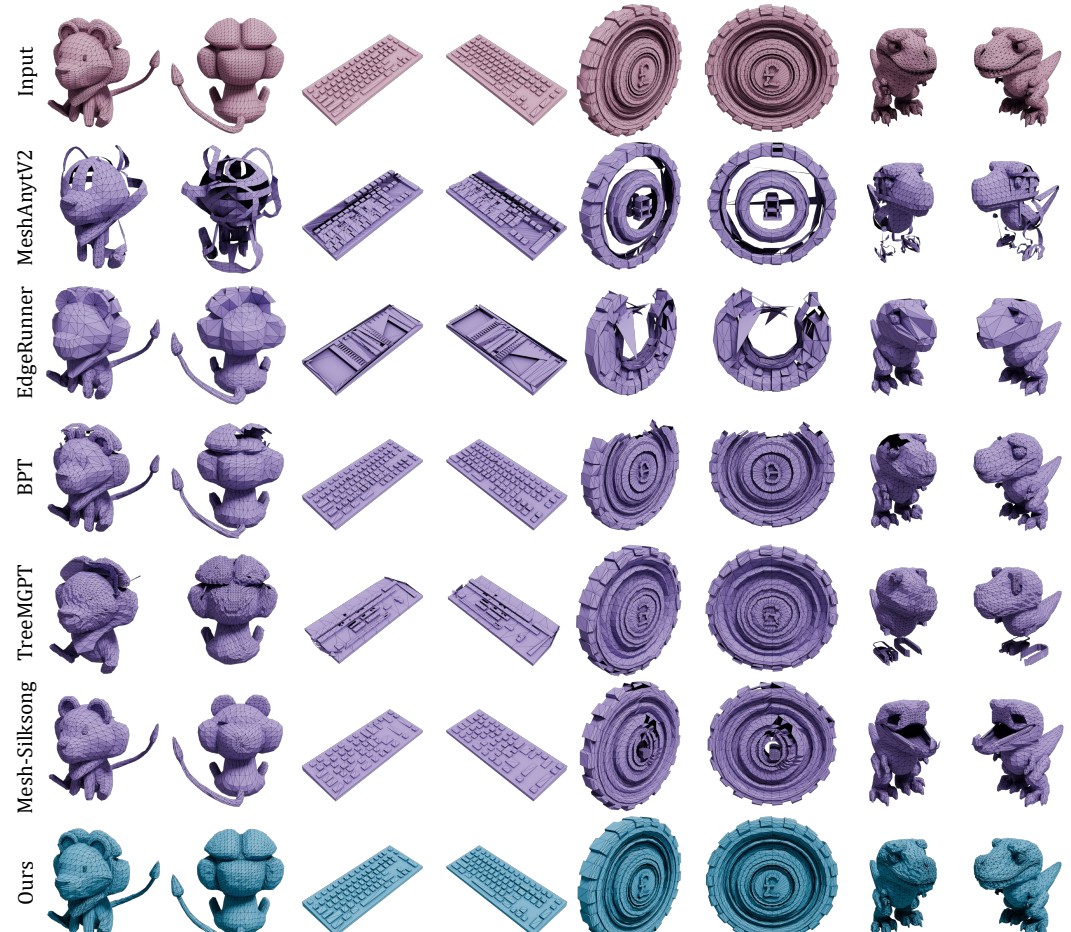

Figure 5: **Qualitative Results on Point-Cloud Conditioned Mesh Generation.**

AdamW(Loshchilov & Hutter, 2019) with cosine-decayed learning rate ($1\times10^{-4} \rightarrow 1\times10^{-5}$), batch size 4 per GPU across 8 GPUs, for 200K steps ( 2 weeks).

**Evaluation Dataset & Metrics.** Prior autoregressive mesh generation works are typically trained on Objaverse, yet the exact subsets used are often unspecified, making replication difficult. To ensure fair comparison, we adopt the Toys4K (Stojanov et al., 2021) dataset containing 4,000 meshes across 105 categories. Generation quality is evaluated with three metrics: Chamfer Distance (CD), which measures the average bidirectional distance between generated and ground-truth point clouds; Hausdorff Distance (HD), which captures the worst-case surface deviation; and Normal Consistency (NC), which assesses the alignment of local surface orientations.

## 4.2 POINT-CLOUD-CONDITIONED MESH GENERATION

To benchmark the performance of point-cloud-conditioned mesh generation, we choose MeshAnythingV2 (Chen et al., 2025b), EdgeRunner (Tang et al., 2025), BPT (Weng et al., 2025), TreeMeshGPT (Lionar et al., 2025), and Mesh-Silksong (Song et al., 2025) as our baselines. We do not compare with Nautilus (Wang et al., 2025b) due to the absence of pretrained checkpoints. During inference, we adopt identical random seed and sampling temperature of 0.5 for all methods.

**Quantitative Results.** Table 1 reports the quantitative evaluation results. Our approach achieves substantial gains over baselines in both CD and HD, indicating that the generated surfaces align more closely with the ground-truth meshes. Moreover, our method attains the highest |NC| and matches the best existing method (Mesh-Silksong) in NC, suggesting the good performance in preserving surface orientation. These advantages stem from the sparse-voxel representation, which provides precise local geometric guidance and allows our model to faithfully reproduce intricate details. In

Table 2: **Comparison on Mesh Tokenization Efficiency.**

| Method | MeshAnythingV2 | EdgeRunner | TreeMeshGPT | BPT | Nautilus | DeepMesh | Mesh-Silksong | Ours |
|---|---|---|---|---|---|---|---|---|
| Compression Ratio ↓ | 0.46 | 0.47 | 0.22 | 0.26 | 0.27 | 0.28 | 0.22 | **0.18** |

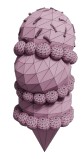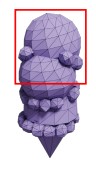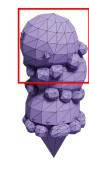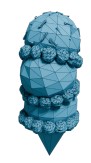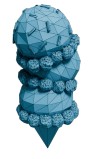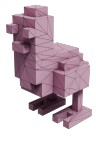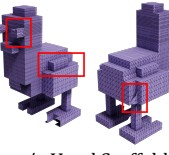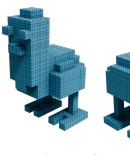

| Input | w/o Voxel Encoder | w/ Voxel Encoder | Input | w/o Voxel Scaffold | w/ Voxel Scaffold |
|---|---|---|---|---|---|

Figure 6: **Qualitative Ablation Studies on Sparse-Voxel Encoder.**

contrast, baseline methods lack such fine-grained supervision, leading to error accumulation, surface drift, and an inability to capture complex local structures. In addition, several prior approaches (e.g., MeshAnythingV2 and EdgeRunner) are constrained by limited tokenization efficiency and therefore train only on meshes with fewer than 4K faces, restricting their capacity to handle more complex geometries. In contrast, our efficient vertex-level tokenization enables training on more complex meshes and raises the performance ceiling.

**Qualitative Results.** Figure 5 visualizes the generated meshes of different methods. It is easy to observe that our method clearly reconstructs finer geometric detail—for example, the key layout of the "keyboard" (second column) and the patern on the coin (third column). Competing methods, while able to capture coarse shape, often suffer from surface misalignment (e.g., "keyboard" in second column), detail loss (e.g., "coin" in third column), or incomplete

Table 1: **Quantitative Results on Point-Cloud-Conditioned Mesh Generation.**

| Method | CD $(\times 10^{-1})\downarrow$ | HD↓ | NC↑ | \|NC\|↑ |
|---|---|---|---|---|
| MeshAnythingV2 | 0.213 | 0.169 | 0.194 | 0.878 |
| EdgeRunner | 0.147 | 0.118 | 0.668 | 0.902 |
| BPT | 0.172 | 0.122 | 0.719 | 0.909 |
| TreeMeshGPT | 0.205 | 0.183 | 0.685 | 0.887 |
| Mesh-Silksong | 0.140 | 0.106 | **0.734** | 0.900 |
| MeshWeaver (ours) | **0.116** | **0.087** | 0.732 | **0.914** |

generation (e.g., MeshAnythingV2 on "dinosaur"). These qualitative results highlight the superiority of our approach in fine-grained mesh generation.

### 4.3 MESH TOKENIZATION

To benchmark the efficiency of mesh tokenization, we compare with both face-traversal-based (Chen et al., 2025b; Tang et al., 2025; Lionar et al., 2025) and coordinate-merging-based (Weng et al., 2025; Wang et al., 2025b; Zhao et al., 2025a; Song et al., 2025) mesh tokenization approaches. We report the mesh compression ratio computed as $L/(9N)$, where $L$ is the compressed sequence length and $9N$ is the sequence length of vanilla representation of a $N$-face mesh, a lower compression ratio indicates better efficiency.

As Table 2 shows, our vertex-level tokenization achieves a state-of-the-art compression ratio of 18%, while existing coordinate-level tokenization algorithms remain capped at about 22%. It is worth noting that the compression efficiency of our tokenization scheme still has room for improvement. For example, during vertex token prediction, one could follow the idea of BPT and adopt separate token sets for patch-center vertices and for other vertices. This design would implicitly distinguish different patches and eliminate the need to insert a BOS token at the beginning of each patch sequence in Equation 3, thereby further shortening the token length. In this work, however, we opt for the simpler implementation of explicitly inserting BOS tokens.

### 4.4 ABLATION STUDIES

**Sparse-Voxel Encoder.** We investigate the contribution of the sparse-voxel encoder from three complementary aspects: (i) *voxel features as vertex representation* (VF), (ii) *cross-attention-guided token prediction* (CA), and (iii) *sparse voxels as generation scaffold* (GS). Among these, VF and CA are part of model training, while GS is used only at inference time. To isolate their effects, we train ablated variants from scratch under identical hyperparameters to the full model. Concretely, without VF we replace voxel features with multi-level static vocabulary embeddings for vertex rep-

resentation; without CA, each level's token prediction head reduces to a linear classifier without cross-attention (please refer to Figure 3); without GS, we disable logit masking based on the sparse-voxel structure during inference.

Quantitative results are reported in Table 3. Removing either VF or CA results in a substantial performance drop, and ablating both leads to the most severe degradation, indicating that voxel-based geometric priors and cross-attention guidance provide complementary benefits for mesh generation. Disabling GS at inference produces a moderate but consistent

Table 3: **Ablation on Sparse-Voxel Encoder.**

| Method | CD ($\times 10^{-1}$)$\downarrow$ | HD$\downarrow$ | NC$\uparrow$ | $|NC|\uparrow$ |
|---|---|---|---|---|
| w/o VF | 0.142 | 0.122 | 0.694 | 0.884 |
| w/o CA | 0.146 | 0.128 | 0.681 | 0.886 |
| w/o VF&CA | 0.158 | 0.138 | 0.660 | 0865 |
| w/o GS | 0.122 | 0.090 | 0.715 | 0.909 |
| Ours | **0.116** | **0.087** | **0.732** | **0.914** |

decline, confirming its role in constraining the generative process around the input surface and mitigating error accumulation and surface drifting, thereby facilitating our "surface weaving" paradigm. We also visualize some qualitative results in Figure 6, where removing the sparse-voxel encoder in training results in detail loss, while disabling the inference-time scaffold results in surface drifting.

**Level Partition.** We study multi-level partition from two perspectives: (i) *space partition*, where the 3D space is divided into multi-level voxel grids with a fixed final resolution of 7-bit quantization (*i.e.*, $2^7 = 128$), and (ii) *layer partition*, which controls how transformer depth is allocated across levels. For space partition, we experiment with three configurations—$(16, 8)$, $(8, 16)$, and $(8, 4, 4)$—that exploit a moderate number of levels while keeping the vocabulary size at each level tractable for training. Other decompositions such as $(32, 4)$ would lead to prohibitively large level 0 vocabulary size (e.g., $32^3$), which would cause training difficulties and inefficiency for cross attention. For layer partition, we fix the total number of self-attention layers at 24 in the autoregressive transformer and vary the allocation across levels; specifically, under the same $(16, 8)$ space partition, we assign the first $M$ layers to level 0 is assigned $M$ layers and the remaining $24-M$ layers to level 1.

As shown in Table 4, the $(16, 8)$ and $(8, 16)$ space partitions achieve comparable performance, while $(8, 4, 4)$ performs a little worse. We attribute this to the deeper hierarchy reducing the spatial support of later levels, limiting the effective range of local geometry injected by sparse-voxel features and thus increases the difficulty of vertex prediction. From an effi-

Table 4: **Ablation Study on Level Partition.**

| Space Part. | Layer Part. | CD ($\times 10^{-1}$)$\downarrow$ | HD$\downarrow$ | NC$\uparrow$ | $|NC|\uparrow$ |
|---|---|---|---|---|---|
| (8,16) | 16+8 | 0.120 | 0.088 | 0.738 | 0.912 |
| (8,4,4) | 16+8 | 0.137 | 0.096 | 0.691 | 0. 880 |
| (16,8) | 18+6 | **0.113** | 0.089 | 0.729 | 0.908 |
| (16,8) | 20+4 | 0.121 | 0.088 | **0.740** | 0.910 |
| (16,8) | 16+8 | 0.116 | **0.087** | 0.732 | **0.914** |

ciency perspective, $(8, 16)$ also requires an extra downsampling layer in the sparse-voxel encoder to match the level 0 resolution, thus we adopt $(16, 8)$ as our default space partition configuration.

Regarding the partition of transformer layers, varying the depth of level 0 from 16 to 20 has negligible effect on final performance. We argue that 16 layers are sufficient to handle the coarse voxel prediction task, and subsequent layers mainly refine predictions at finer levels, which does not require excessive depth. In our final model, we choose a 16+8 split for level 0 and level 1, respectively.

**Cross-Attention KV Cache.** We further evaluate the effect of the cross-attention KV cache introduced in Section 3.4 on inference efficiency. Specifically, we randomly select 200 meshes from the Toys4K dataset and measure the throughput of point-cloud-conditioned mesh generation on the same GPU, quantified by the number of generated tokens per second (tokens/s). Without cross-attention KV caching, the model runs at average speed of 26.8 tokens/s, while enabling the cache increases the throughput to 30.7 tokens/s—an improvement of approximately 14.5%.

## 5 CONCLUSION

We introduced MeshWeaver, an autoregressive framework that casts mesh generation as a surface weaving process. By predicting the next vertex rather than the next coordinate and coupling decoding with a hierarchical sparse-voxel encoder, our model achieves shorter sequences, stronger structural reasoning, and fine-grained geometric guidance. This design enables state-of-the-art compression and geometric fidelity while scaling to meshes with up to 16K faces. Beyond these results, MeshWeaver suggests a promising path toward practical, high-quality mesh generators that tightly unite structural coherence with geometric detail.

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

## A IMPLEMENTATION DETAILS

**Point Cloud Conditioner.** Similar to prior works, we adopt a point cloud encoder based on the architecture of 3DShape2VecSet (Zhang et al., 2023) to encode the input point cloud into fix-length conditional tokens, which are prepended to the mesh sequence to provide global generation context. The conditional tokens are attended to each other via bidirectional attention, while the subsequent vertex tokens attend to their predecessors via causal attention. This attention mechanism can be implemented efficiently with PyTorch's (Paszke et al., 2019) FlexAttention.

**Training Loss.** Our framework adheres to a fully causal generation scheme: each vertex is conditioned on all preceding vertices, and within a vertex, each level's token is conditioned on predictions from coarser levels. Training therefore reduces to a sequence-modeling problem, optimized using the standard cross-entropy loss commonly employed in causal language models:

$$\mathcal{L} = -\sum_{j=1}^{N} \sum_{l=0}^{L-1} \log p\big(v_j^l \mid \boldsymbol{v}_{<j}, v_j^{<l}\big), \tag{6}$$

where $N$ denotes the number of vertices, $L$ the number of levels, $v_j^l$ the level-$l$ token of vertex $j$, $\boldsymbol{v}_{<j}$ all previously generated vertices, and $v_j^{<l}$ the already predicted levels of the $j$-th vertex.

## B ADDITIONAL RESULTS

**Traning Curve Ablation on Sparse-Voxel Encoder.** We further visualize training dynamics in Figure 7. Compared to the ablated variant without VF and CA, our full model exhibits markedly faster convergence. Without the sparse-voxel encoder, the model effectively reduces to pure language modeling over vertices, lacking geometric priors and therefore requiring substantially longer optimization. In contrast, the inclusion of the sparse-voxel encoder provides explicit surface-aware context, reducing the difficulty of next-vertex prediction and accelerating training.

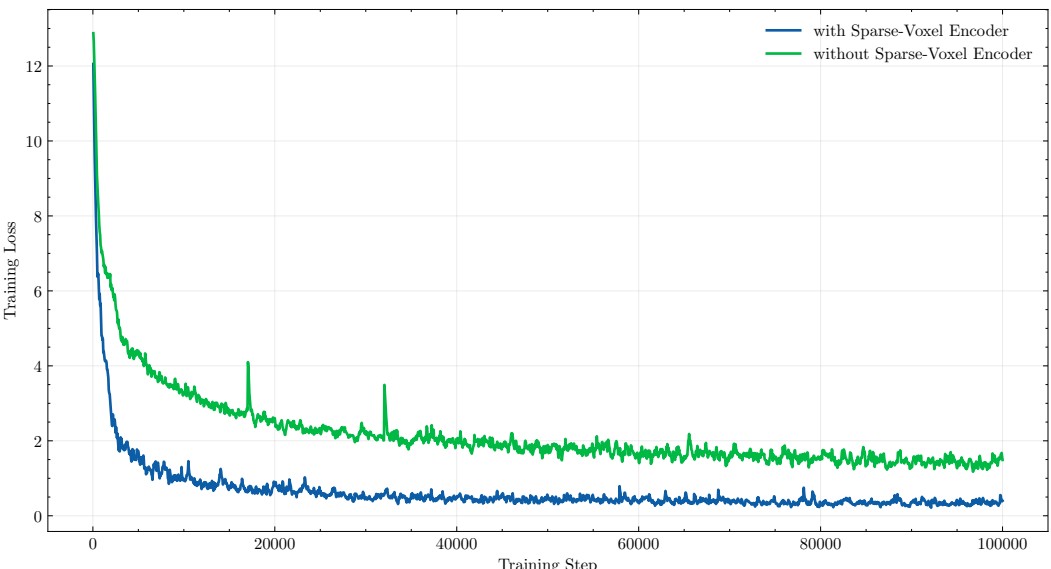

Figure 7: **Comparision on Training Loss with or without Sparse-Voxel Encoder.**

**Additional Qualitative Results.** We present more generated results in Figure 8 and Figure 9.

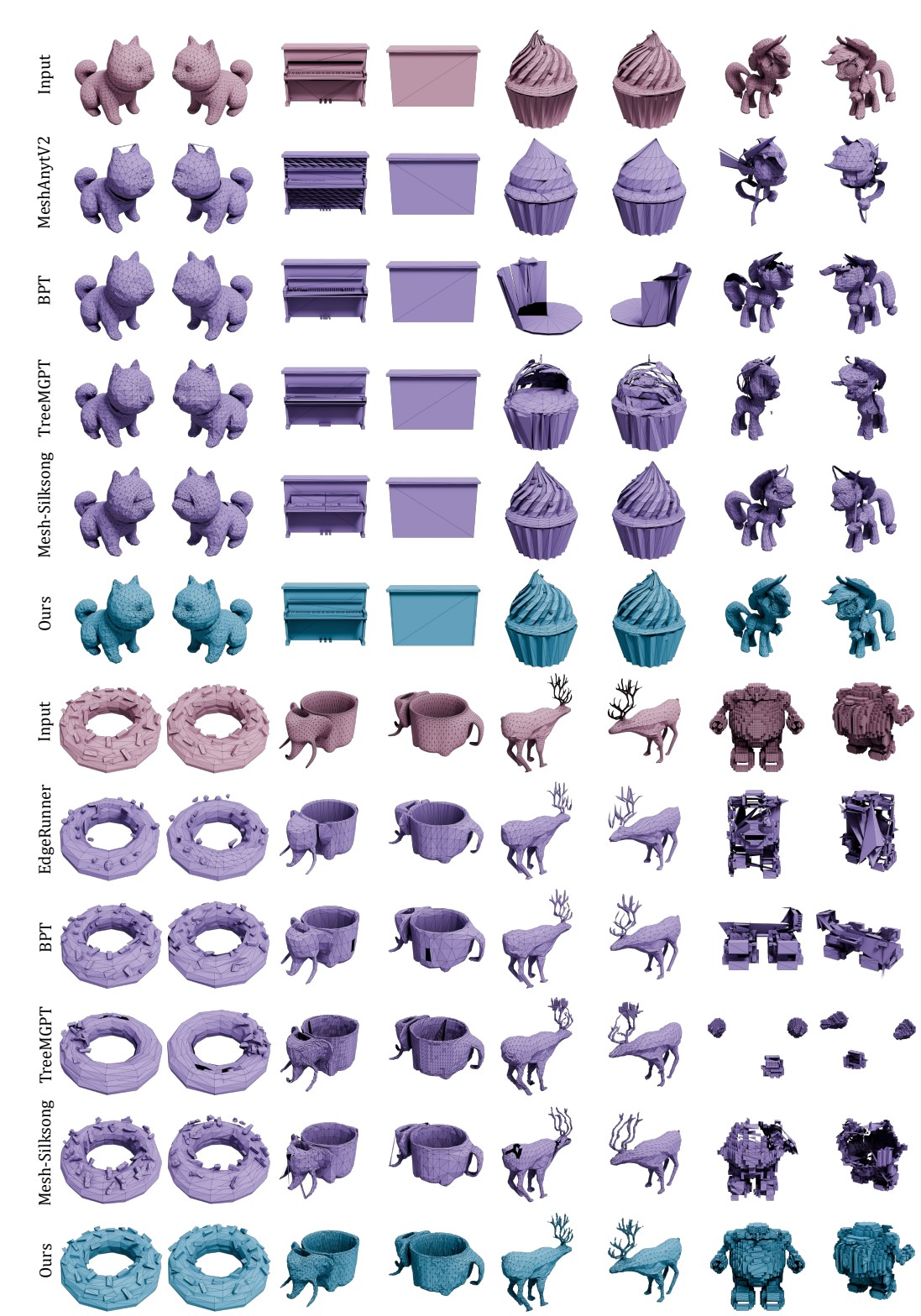

Figure 8: **Additional Results on Point-Cloud Conditioned Mesh Generation.**

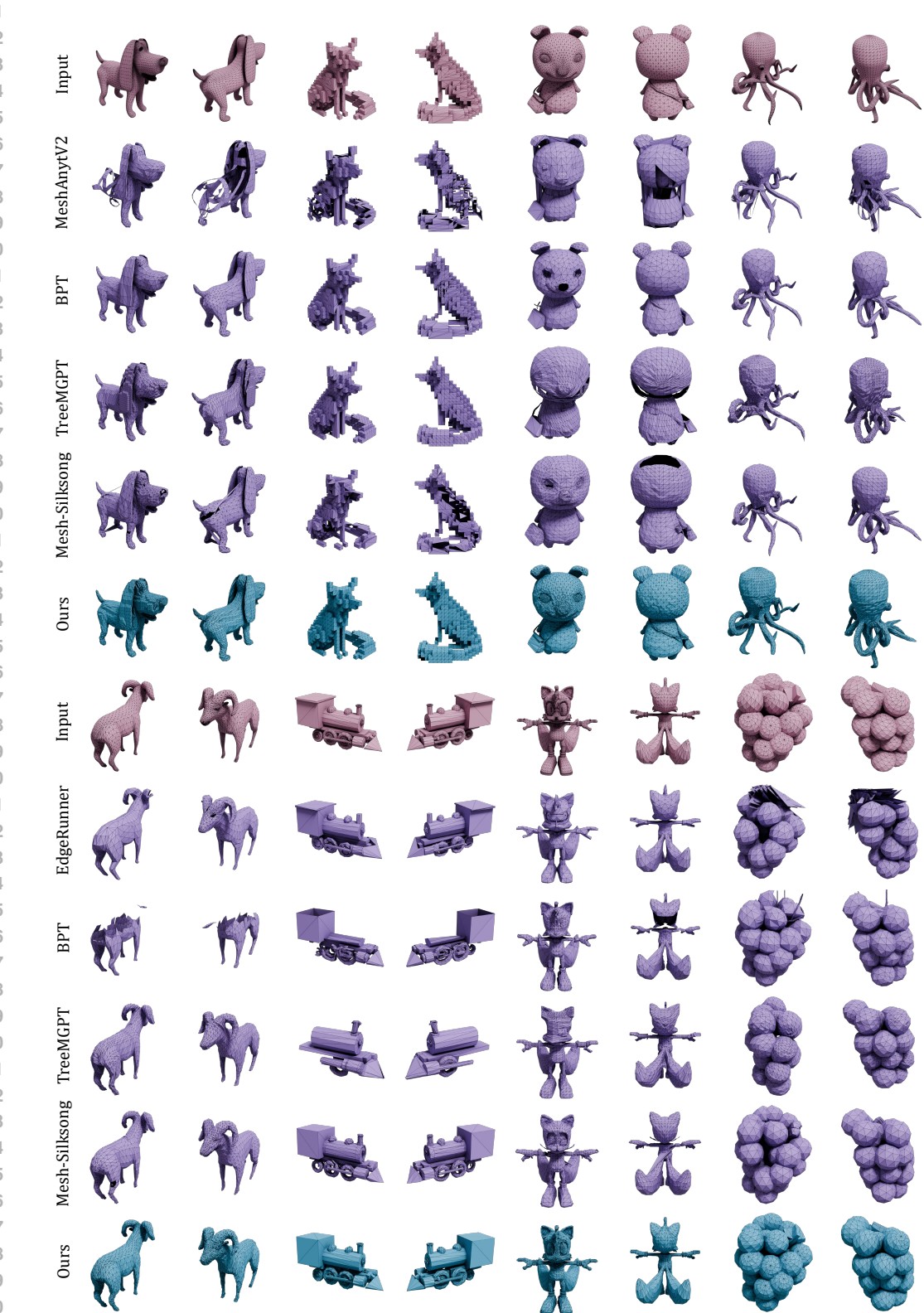

Figure 9: **Additional Results on Point-Cloud Conditioned Mesh Generation.**

## C  LIMITATIONS

While MeshWeaver advances the state of automatic mesh generation, several challenges remain. First, real-world assets often contain tens to hundreds of thousands of faces, which are still beyond the capacity our framework can reliably produce. Second, the sparse-voxel encoder, though effective for structural guidance, introduces additional computational overhead, making it difficult to scale to very high resolutions (e.g., $512^3$ or $1024^3$). Finally, performance is bounded by the scale and quality of available training data; we expect that larger and more diverse curated datasets will further improve both fidelity and robustness.

