# OpenReview forum: "MeshWeaver: Sparse-Voxel-Guided Surface Weaving for Autoregressive Mesh Generation"
_ICLR.cc/2026/Conference — ICLR 2026 Conference Withdrawn Submission_

### Official Review · Reviewer_Jv2K · 2025-10-18

**Soundness:** 2
**Presentation:** 3
**Contribution:** 2
**Rating:** 4
**Confidence:** 5

**Summary:**

This paper introduces MeshWeaver, an autoregressive framework for mesh generation that aims to solve two key limitations of prior work: low tokenization efficiency and a lack of local geometric guidance. The method proposes two main contributions. First, it reformulates the generation task from next-coordinate to next-vertex prediction, where a complete vertex is predicted in a multi-level, coarse-to-fine manner in a single decoding step, achieving a higher mesh compression ratio. Second, it introduces a hierarchical sparse-voxel encoder to inject fine-grained geometric context into the transformer at three levels: as a feature representation for vertices, as a key-value source for cross-attention, and as a spatial scaffold to constrain predictions. Experiments show that MeshWeaver achieves a state-of-the-art compression ratio, scales to 16K-face meshes, and improves geometric fidelity over selected baselines.

**Strengths:**

1. The shift from predicting individual coordinates to predicting a complete vertex via a multi-level hierarchical token is a novel and conceptually interesting approach to mesh tokenization. It achieves a significantly better compression ratio compared to prior methods.
2. The design of the sparse-voxel encoder and its three-pronged integration into the generation process (representation, attention, scaffolding) is a technically sound and well-engineered system. It demonstrates a clear effort to tightly couple the generative model with the underlying 3D geometry.
3. The quantitative and qualitative results presented in the paper show a clear improvement in geometric fidelity (CD, HD, NC) over the compared methods like BPT and MeshAnythingV2. The visual examples effectively highlight the model's ability to capture finer details.

**Weaknesses:**

1. The proposed framework appears fundamentally unscalable for practical, high-quality applications. The reported **7-bit (128^3) coordinate resolution is entirely insufficient** for any industrial use case and represents a major limitation. The choice of a plain, monolithic transformer is computationally heavy and ill-suited for this task compared to more modern and efficient alternatives.
2.  The paper's core premise—that extreme sequence compression via next-vertex prediction is a desirable path forward—is fundamentally flawed. This approach **drastically increases the learning difficulty** of the sequence modeling task. Forcing the transformer to predict a complex, multi-level vertex token in one step is a much harder problem than predicting a simple coordinate. This detrimental trade-off likely restricts the model's ability to learn complex topological and geometric relationships, ultimately limiting the quality ceiling of the meshes it can generate. The pursuit of compression in this manner seems to be a counter-productive research direction.
3. The paper fails to compare against state-of-the-art models that employ more suitable architectural solutions. Architectures like **Hourglass transformers with sliding windows** can effectively handle long sequences for high-face-count meshes, directly challenging the paper's motivation for aggressive compression. Furthermore, by not comparing to specialized re-topology methods designed for AI-generated assets, the paper's practical utility remains unproven. The claims of superiority are not sufficiently validated without these critical baselines.
4. Collectively, the issues of low resolution, an inefficient backbone architecture, and a potentially misguided focus on compression over learnability suggest the proposed method offers **no clear contribution toward making mesh generation industrially viable**. A critical real-world application is the clean re-topology of dense meshes from other generative AI systems, a scenario where this model is expected to perform poorly.

**Questions:**

1. The reported 7-bit (128^3) coordinate quantization is insufficient for high-quality results. Could the authors conduct an experiment using a **10-bit (1024^3) resolution**? We suspect the multi-level vertex prediction scheme would become intractable due to the massive vocabulary size at the coarse level, and this experiment would directly test the true scalability of the proposed tokenization.
2. Why was a standard transformer used instead of a more efficient architecture like an **Hourglass transformer**, which is better suited for hierarchical spatial data? Could you provide an ablation study comparing the performance, scalability, and parameter count against such a baseline?
3.  How do the authors justify that the increased learning difficulty of predicting a multi-level vertex token is a worthwhile trade-off for a better compression ratio, especially when alternative architectures can handle longer sequences directly? What evidence suggests this is a more promising research direction than improving coordinate-based models?
4. A critical industrial use case for mesh generation is the re-topology of dense, often noisy, outputs from other generative models. How does MeshWeaver perform in this scenario? We suggest a direct comparison against specialized methods like **Tripo/Hunyuan3D's PolyGen on dense meshes** to assess its practical utility.

---

### Official Review · Reviewer_7L5U · 2025-10-25

**Soundness:** 2
**Presentation:** 3
**Contribution:** 2
**Rating:** 4
**Confidence:** 4

**Summary:**

This paper proposes MeshWeaver for autoregressive 3D mesh generation, utilizing a sparse-voxel-guided surface weaving process. The core idea involves exploring a new type of condition encoder based on sparse voxels, combined with a 2D vertex-token compression scheme to effectively convert a mesh into a sequence for a language-modeling-style approach. The authors demonstrate that this combination achieves enhanced performance, yielding high-quality meshes and a promising mesh compression ratio, supported by a detailed set of experimental results.

**Strengths:**

- The work introduces and explores a new type of condition encoder, the sparse voxel encoder, for the mesh generation task, and it successfully demonstrates enhanced performance.
- The experimental results are promising, and the authors provide detailed ablation study to investigate various aspects of their model.
- The 2D vector-token tokenization approach also yields an impressive state-of-the-art mesh compression ratio.

**Weaknesses:**

- The primary experimental limitation is the relatively low generation resolution of 256, which makes the comparison against recent state-of-the-art methods and the scalability to higher resolution less convincing.
- The ablation study currently still mixes the contributions of the two main components: the sparse voxel encoder and the 2D vertex-token compression. To properly assess their individual impacts, a separate experiment such as using a previous point cloud based encoder alongside the proposed tokenization algorithm can be helpful. The current ablation setup (no VF & CA) does not offer a clear comparison to prior works.
- Additionally, the paper lacks an analysis of failure cases, which would be highly beneficial for evaluating the method’s generalizability and robustness.

**Questions:**

- How is the 18% compression ratio calculated exactly? A simple example to illustrate the tokenization idea (with a specific mesh and hyper-parameters) would be very helpful to clarify the mechanism and the difference compared to BPT.
- What are the values of $L$ and $D$ used in the experiments? It seems $L$ is only 2 according to the figures; have the authors tried to use other numbers of levels for $L$ to explore the trade-off?

---

### Official Review · Reviewer_x4Gf · 2025-10-31

**Soundness:** 3
**Presentation:** 3
**Contribution:** 3
**Rating:** 6
**Confidence:** 4

**Summary:**

The paper presents MeshWeaver, an auto-regressive mesh generator via next vertex prediction. For geometry conditioned generation, the proposed method adopts a hierarchical approach to progressively narrow the cross-attention region to restrict the generated vertex to fit to the surface. The method achieves superior performance than previous methods with a shorter sequence and better performance.

**Strengths:**

- The paper presents a state-of-the-art auto-regressive mesh generation method designed specially for geometry conditioned mesh generation, *e.g.* the geometry cross attention, and the vertex representations.

- The proposed method achieves a state-of-the-art mesh compression ratio of 18%, outperforming traditional coordinate-level approaches capped at 22%.

- The proposed method achieves superior generation performance than prior arts.

**Weaknesses:**

- Since the cross-attention region is selected by layer-wise prediction, there are accumulation of errors during inference, and could pose more severe training / testing gap than naive full cross-attention.

- Similar to tricks in detection transformers, each decoder layer is asked to perform detection (Eq. 6). I am curious whether and where this trick is activated while performing ablations in Table 3. Additionally, would this trick alone improves the generation performance?

**Questions:**

See weakness part.

---

### Note · Authors · 2025-11-13

I have read and agree with the venue's withdrawal policy on behalf of myself and my co-authors.